



# PSCs initiated by mountain waves in a global chemistry-climate model: A missing piece in fully modelling polar stratospheric ozone depletion

Andrew Orr[1], J. Scott Hosking[1], Aymeric Delon[2], Lars Hoffman[3], Reinhold Spang[4], Tracy Moffat-Griffin[1], James Keeble[5,6], Nathan Luke Abraham[5,6], and Peter Braesicke[7]

[1]British Antarctic Survey, Cambridge, UK
[2]Ecole normale supérieure Paris-Saclay, Paris, France
[3]Forschungszentrum Jülich, Jülich Supercomputing Centre, Jülich, Germany
[4]Forschungszentrum Jülich, Institut für Energie und Klimaforschung, Stratosphäre, IEK-7, Jülich, Germany
[5]National Centre for Atmospheric Science (NCAS), University of Cambridge, Cambridge, UK
[6]Department of Chemistry, University of Cambridge, Cambridge, UK
[7]Karlsruher Institut für Technologie, Institut für Meteorologie und Klimaforschung, Karlsruhe, Germany

*Correspondence to*: Andrew Orr (anmcr@bas.ac.uk)

**Abstract.** An important source of polar stratospheric clouds (PSCs), which play a crucial role in controlling polar stratospheric ozone depletion, is from the temperature fluctuations induced by mountain waves. These enable stratospheric temperatures to fall below the threshold value for PSC formation in regions of negative temperature perturbations or cooling-phases induced by the waves even if the synoptic-scale temperatures are too high. However, this formation mechanism is usually missing in global chemistry–climate models because these temperature fluctuations are neither resolved nor parameterised. Here, we investigate in detail the episodic and localised wintertime stratospheric cooling events produced over the Antarctic Peninsula by a parameterisation of mountain-wave-induced temperature fluctuations inserted into a 30-year run of the global chemistry-climate configuration of the UM-UKCA (Unified Model - United Kingdom Chemistry and Aerosol) model. Comparison of the probability distribution of the parameterised cooling-phases with those derived from climatologies of satellite-derived AIRS brightness temperature measurements and high-resolution radiosonde temperature soundings from Rothera Research Station on the Antarctic Peninsula shows that they broadly agree with the AIRS-observations and agree well with the radiosonde-observations, particularly in both cases for the "cold tails" of the distributions. It is further shown that adding the parameterised cooling-phase to the resolved/synoptic-scale temperatures in the UM-UKCA model results in a considerable increase in the number of instances when minimum temperatures fall below the formation temperature for PSCs made from ice water during late austral autumn / early austral winter and early austral spring, and without the additional cooling-phase the ice frost point is rarely exceeded above the Antarctic Peninsula in the model. Similarly, it was found that the formation potential for PSCs made from ice water was many times larger if the additional cooling is included. For PSCs made from NAT particles it was only during October that the additional cooling is required for the NAT temperature threshold to be exceeded





(despite more NAT PSCs occurring during other months). The additional cooling-phases also resulted in an increase in the
surface area density of NAT particles throughout the winter and early spring, which is important for chlorine activation. The
parameterisation scheme was finally shown to make substantial differences to the distribution of total column ozone during
October, resulting from a shift in the position of the polar vortex.

# 1 Introduction

Polar stratospheric clouds (PSCs) are important in polar ozone chemistry as reactions on their surfaces convert reservoir species
into highly reactive ozone-destroying gases containing chlorine and bromine, which contributes to the depletion of the
Antarctic and Arctic stratospheric ozone layer (Solomon, 1999). The ozone destruction is further aided by the removal of nitric
acid via the sedimentation of nitric acid containing PSCs (so-called denitrification), which reduces the deactivation of active
chlorine (Fahey et al., 1990). These recurring processes have resulted in the severe stratospheric ozone-depletion over Antarctic
during springtime in recent decades, commonly referred to as the "ozone hole" (Farman et al., 1985; Solomon et al., 1986),
which has resulted in considerable changes in the Southern Hemisphere circulation (e.g. Thompson and Solomon, 2002; Orr
et al., 2008; Polvani et al., 2011).

One of the main requirements for PSCs to form is for very cold stratospheric temperatures, which are lower than some
minimum threshold $T_{NAT}$ for PSCs consisting of nitric acid trihydrate (NAT) particles, $T_{STS}$ for PSCs consisting of liquid
supercooled ternary solutions (STS), and $T_{ice}$ for PSCs consisting of water ice particles. At an altitude of around 20 km the
threshold temperatures are generally assumed to be around 195 K for $T_{NAT}$, 191 K for $T_{STS}$, and 188 K for $T_{ice}$ - although these
can vary as they are also dependent on the amounts of gases such as nitric acid and water vapour (Pawson et al., 1995; Alfred
et al., 2007).

In the Antarctic winter, temperatures are often low enough in the stratosphere to drop below the threshold temperatures,
resulting in the formation of PSCs over large regions and for extended periods of time (Campbell and Sassen, 2008). However,
if synoptic-scale temperatures are too high for the formation of PSCs, as can occur for example over the edge-region of the
Antarctic stratospheric vortex (particularly during early winter and early spring), the addition of negative temperature
anomalies induced by vertically propagating wave motion forced by stratified flow over high mountains can result in
temperatures falling below the thresholds for PSC formation, i.e. the formation of PSCs due to mountain wave activity
(Alexander et al., 2011, 2013; Orr et al., 2015). Hereafter, these localised negative temperature anomalies, which form in the
upwelling portion of the wave through adiabatic expansion, will be referred to as the "cooling-phase" of mountain waves. In
the Arctic, because it is synoptically warmer than the Antarctic due to disturbances from transient planetary waves, this
mechanism is especially important for the formation of PSCs (Dörnbrack and Leutbecher, 2001; Alexander et al., 2013).
Regions known to be a source of remarkable mountain-wave-induced stratospheric cooling that can trigger the formation of
PSCs include the Antarctic Peninsula, Scandinavia, and Greenland (Dörnbrack et al., 1999, 2002, 2012; Alexander and
Teitelbaum, 2007; Plougonven et al., 2008; Eckermann et al., 2009; Noel et al., 2009; Hoffman et al., 2013, 2016, 2017).





However, mountain-wave-induced PSC formation (and associated ozone depletion) is missing from current global chemistry-climate models. This is because they are unable to explicitly resolve localised mountain-wave dynamics and their associated temperature perturbations due to their coarse spatial resolution, which is on the order of a few hundred kilometres, while mountain waves typically have a wavelength of around 100 km or smaller. This failure was addressed in previous work
described by Orr et al. (2015), which inserted a parameterisation scheme describing stratospheric mountain-wave-induced temperature fluctuations into the UM-UKCA global chemistry-climate model, consisting of version 7.3 of the HadGEM3 (Hadley Centre Global Environment Model version 3) global climate model configuration of the Unified Model (UM) (Hewitt et al., 2009), coupled to the United Kingdom Chemistry and Aerosol (UKCA) module (Morgenstern et al., 2009). This work showed that the parameterised temperature fluctuations over the Antarctic Peninsula were broadly in agreement with detailed
results using a high-resolution regional climate model, and also that the amount of PSCs simulated over the Antarctic Peninsula by the chemistry-climate model  increased considerable following the inclusion of the cooling-phase of the parameterised temperature fluctuations. Novel developments such as this that make global chemistry-climate models more physically based / comprehensive are needed to improve our ability to make accurate predictions of stratospheric ozone, especially related to the expected recovery of the Antarctic ozone hole by approximately mid-century (and its role in offsetting the effects of
increasing greenhouse gases), which requires the use of interactive stratospheric ozone chemistry for projections (Chiodo and Polvani, 2017; Pope et al., 2020). The recovery of stratospheric ozone levels (together with greenhouse gas increases) is expected to result in profound changes to the high-latitude Southern Hemisphere climate system, primarily by affecting both the strength and latitude of the westerly polar jet (Eyring et al., 2013; Previdi and Polvani, 2014; Iglesias-Suarez et al. 2016; Chiodo and Polvani, 2017).

This study further investigates the parameterised mountain-wave-induced cooling-phase computed by the UM-UKCA model described in Orr et al. (2015), focusing particularly on its rigorous validation to better constrain the scheme and an assessment of its impact on the formation potential (FP) of PSCs (Dörnbrack and Leutbecher, 2001), which is necessary before any assessment of the global impact on polar ozone chemistry. The investigation will again primarily focus on the Antarctic Peninsula due to it being a hot-spot for mountain-wave-induced PSCs in Antarctica and thus a highly suitable test-case, and
will examine locally: i) a comparison of the distribution of observed and parameterised mountain-wave-induced stratospheric cooling-phase, ii) the impact of the parameterisation scheme on minimum temperatures and the FP of PSCs, iii) an investigation into the conditions that produce mountain-wave-induced stratospheric cooling in the parameterisation scheme, and iv) the impact of the scheme on local PSC formation and heterogeneous chemistry. The investigation will finish by investigating the non-local impacts of the scheme by examining changes to ozone as well as temperature and pressure over the high-latitude
Southern Hemisphere.





## 2 Materials and methods

### 2.1 Description of parameterisation scheme and inclusion in global chemistry-climate model

The mountain wave scheme is described by Dean et al. (2007) and computes the maximum negative $\Delta T_{SSO}^-$ and positive $\Delta T_{SSO}^+$ temperature fluctuations associated with the positive and negative vertical parcel displacement of gravity waves generated by

flow passing over subgrid-scale orography (SSO) in a climate or general circulation model. The approach assumes that the vertical propagation is described by linear hydrostatic mountain waves, generated by steady-state stratified flow over an isolated two-dimensional ridge, i.e. in the absence of wave dissipation mechanisms the change in wave amplitude/displacement with height is controlled by variations in the air density, the horizontal wind speed $U$ (resolved in the direction of the wave vector), and the buoyancy frequency $N$. The scheme includes critical-level absorption and wave breaking to prevent the wave

amplitude from exceeding the local "saturation amplitude", defined as $U/NF_{sat}$ (where $F_{sat}$ is the critical Froude number for saturation). The initial wave amplitude is set equal to the "effective" mountain height $h_{eff}$ of the SSO (i.e. $h - h_b$, where $h = n_\sigma \sigma$ is the height of the SSO and $h_b = h - U_0/N_0 F_C$ is the height of the blocked layer, $\sigma$ is the standard deviation of the SSO, $n_\sigma$ is a constant, $F_C$ is the critical Froude number at which flow blocking is deemed to first occur, and the subscript "0" refers to quantities averaged between the ground and $h$).

As mentioned above, the scheme was previously inserted into the UM-UKCA global chemistry-climate model. UM-UKCA uses a quasi-equilibrium PSC scheme which models two types of PSC particles: NAT and mixed NAT/ice, both calculated assuming thermodynamic equilibrium with gas-phase HNO3 and H2O (following Chipperfield, 1999). For NAT particles, the saturation vapour pressure of HNO3, calculated following Hanson and Mauersberger (1988), is used to calculate the mass of HNO3 in the solid phase, while for mixed NAT/ice the saturation vapour pressure of water vapour over ice is calculated

following Goff (1957). Surface area density for both PSC types are calculated assuming spherical particles with fixed density and radii. For NAT particles these are 1.35 g cm⁻³ and 1 μm, and for mixed NAT/ice particles 0.928 g cm⁻³ and 10 μm, respectively. As a result, in this scheme each individual NAT or mixed NAT/ice particle is assumed to be the same size, while the number density, and so surface area density, changes with the availability of HNO3 and H2O, as well as temperature and pressure.

Only the cooling-phase $\Delta T_{SSO}^-$ computed by the mountain wave scheme is coupled / passed to the PSC scheme, i.e. the PSC scheme uses as input a "total" temperature $T = T_{UM-UKCA} + \Delta T_{SSO}^-$, where $T_{UM-UKCA}$ is the temperature explicitly resolved by the UM-UKCA model. The cooling-phase only is used because, in the simple quasi-equilibrium PSC scheme, an instantaneous temperature rise will evaporate particles immediately if the temperature increases above the PSC formation threshold - when in reality this would take some time. This configuration - referred to from now on as the "perturbation" simulation - was run

for 30 years (following a spin-up period of 30 years) for a perpetual year 2000 at a horizontal resolution of N48 (equivalent to a grid spacing of 2.5º × 3.5º) and 60 vertical levels (up to 84 km), using prescribed sea-ice fraction and sea surface temperature. Note that values of the constants/parameters used by the scheme were set to $n_\sigma = 3$, $F_{sat} = 2$, and $F_C = 4$, which were selected following initial analysis to optimize its performance over the Antarctic Peninsula by best matching the magnitude of





the parameterised stratospheric temperature fluctuations with those explicitly resolved by a high-resolution regional

configuration of the UM (see Orr et al. (2015) for further details). A control experiment - referred to from now on as the

"control" simulation - was also run, which is identical to the perturbation run but with the exception that the mountain wave

scheme is switched off. Orr et al. (2015) provides more details of both experiments. Output from both the model runs are at 6-

hourly intervals (including values of $\Delta T_{SSO}^{-}$ from the perturbation run) and are used as the basis for all subsequent analysis.

Note that earlier studies such as Orr et al. (2012) and Keeble et al. (2014) show that this model well represents the high-latitude

Southern Hemisphere circulation and temperature structure. Nevertheless, of especial importance is an accurate representation

of circumpolar westerly flow at a height of say 850 hPa because of its role in generating wave activity over the Antarctic

Peninsula (Orr et al., 2008). To test this here, the 30-year mean wind at 850 hPa for austral winter (June-July-August) from

the control experiment was compared to the climatological mean from the reanalysis-product ERA5 (i.e. the fifth-generation

reanalysis product from ECMWF, Hersbach and Dee, 2016) over the 1979 to 2019 period, and shown to be in excellent

agreement (not shown).

## 2.2 Data

We use estimates of the amplitude of mountain-wave-induced cooling (i.e. maximum cooling) from Atmospheric Infrared

Sounder (AIRS) measurements of radiance perturbations for a 16-yr period from 2002 to 2017 (Hoffmann et al., 2016, 2017),

as well as from radiosonde soundings for a 14-yr period from 2002 to 2015 (Moffat-Griffin et al., 2011). The nadir scanning

AIRS instrument is on board NASA's Aqua satellite, which since 2002 has typically made four passes per day over the

Antarctic Peninsula, performing an across-track scan covering a distance of 1765 km on the ground. Each scan consists of 90

individual footprints that vary in size between $13.5 \times 13.5$ km at nadir and $41 \times 21.4$ km at the scan edges. Here we use the

666.5 cm$^{-1}$ radiance channel of AIRS, which peaks in sensitivity to atmospheric temperatures at an altitude of around 22 km

and has a full width at half maximum of 9 km, i.e. encompassing an altitude range that is particularly favourable for the

formation of PSCs. See Figure 1 from Orr et al. (2015) for a plot showing the temperature weighting function for this channel.

The minimum radiance perturbation values (i.e. maximum cooling) are calculated for each single footprint. Note that the

relatively coarse vertical resolution of AIRS limits the detection of waves with vertical wavelengths less than approximately

12 km, resulting in the attenuation of the measured wave amplitude, i.e. AIRS underestimates the true wave amplitude at short

vertical wavelengths (Hoffman et al., 2017). Note also that AIRS observes temperature disturbances from both orographic and

155    non-orographic source regions, which in the context of this study would include those generated by storms over the Drake

Passage to the north of the Antarctic Peninsula (Plougonven et al., 2012). The radiosonde soundings were launched around

two to four times per week from Rothera Research Station, which is located along the western side of the Antarctic Peninsula.

See Moffat-Griffin et al. (2011) for more details of the soundings. Figure 1 shows a map of the Antarctic Peninsula, which

includes the location of Rothera Research Station, as well as orography from the Bedmap2 dataset (Fretwell et al., 2013).



## 2.3 Methodology

To verify the parameterised mountain-wave-induced stratospheric cooling-phase, 6-hourly values of $\Delta T_{SSO}^{-}$ for May-June-July-August-September-October over the Antarctic Peninsula from the perturbation run were compared against brightness temperature fluctuations measured by AIRS and temperature fluctuations measured by the radiosonde soundings. The brightness temperature fluctuations measured by AIRS are determined by removing a fourth-order polynomial function, representing the background atmosphere, from the original brightness temperatures (see Orr et al., 2015). To facilitate a comparison with the AIRS-observed minimum brightness temperature fluctuations ($\Delta BT_{AIRS}^{-}$) over the Antarctic Peninsula, the values of $\Delta T_{SSO}^{-}$ are converted into brightness temperature ($\Delta BT_{SSO}^{-}$) by computing a weighted-sum of $\Delta T_{SSO}^{-}$ over all vertical model levels from 15 to 45 km, i.e. by summing the value of $\Delta T_{SSO}^{-}$ multiplied by the associated normalised weighting function for the 666.5 cm$^{-1}$ radiance channel of AIRS over the range of vertical levels. Figure 1 shows the region over the Antarctic Peninsula that were used to compute $\Delta BT_{AIRS}^{-}$ and $\Delta BT_{SSO}^{-}$. Note that the weighting function of the 666.5 cm$^{-1}$ radiance channel is largely insensitive to atmospheric temperatures at altitudes both above 45 km and below 15 km. For the radiosonde-based measurements, we focus on the temperature perturbations $\Delta T_{RS}^{-}$ observed at an altitude of between 20.2 and 20.6 km above sea level (chosen because this range is both in the lower stratosphere and includes the vertical level of the UM-UKCA model at a height of 20.4 km for comparison), which are computed by removing a third-order polynomial function representing the background atmosphere from the original profile (see Moffat-Griffin et al., 2011). The distributions for the parameterised fluctuations are compared with those for the measurements, and the probability density functions generated using a kernel density estimation.

We use output from the two simulations to examine the temperature distribution $T - T_{NAT}$ and $T - T_{ice}$, where $T$ is equal to either $T_{UM-UKCA} + \Delta T_{SSO}^{-}$ (as used by the perturbation run) or $T_{UM-UKCA}$ (as used by the control run), and $T_{NAT}$ and $T_{ice}$ are the actual threshold temperatures for the existence of PSCs composed of NAT and water ice particles, respectively. The values computed for $T_{NAT}$ and $T_{ice}$ are sensitive to the temperature, pressure, HNO$_3$, and water vapour mixing ratio (Hansen and Mauersberger, 1988; Marti and Mauersberger, 1993), which are taken from either the perturbation or control runs. We also compute for each run the FP of PSCs at an altitude of around 20.4 km, using a metric which depends on both the size of the temperature difference below either $T_{NAT}$ or $T_{ice}$, as well as the area of the region. For example, the FP for PSCs composed of NAT particles would be defined as:

$$FP_{NAT} = \begin{cases} 0 & , T - T_{NAT} > -0.1\ K \\ \sum_{i=1}^{N} A_i (T - T_{NAT})_i & , T - T_{NAT} \leq -0.1\ K \end{cases} \qquad \text{Eq. (1)}$$

where $i$ is an integer, $N$ is the total number of model grid-boxes within the region defined in Fig. 1, and $A_i$ is the spatial area of the model grid-box. An analogous equation exists for the FP for PSCs composed of water ice. Note that as the latitude/longitude grid used by the UM-UKCA model has non-uniform spacing / grid-box area (due to varying longitude), the



results from Eq. (1) are also scaled by the cosine of latitude. Note also that again the results are computed for the box situated over the Antarctic Peninsula shown in Fig. 1.

To identify the role of atmospheric conditions on controlling the parameterised stratospheric temperature fluctuations, the sensitivity of the amplitude of the cooling-phase $\Delta T_{SSO}^-$ to the vertical wind shear α is examined, with:

$$\alpha = \frac{U(z_2) - U(z_1)}{U(z_1)} \qquad \text{Eq. (2)}$$

where $z_1 = 0.85$ km, $z_2 = 21.0$ km, and here $U$ is the zonal wind velocity (applicable because the large-scale wind regime over the region containing the Antarctic Peninsula is predominately zonal (Thompson and Wallace, 2000)). Additionally, the sensitivity of $\Delta T_{SSO}^-$ to directional shear was also investigated by examining its relationship to a change in the direction of the wind with height, between $z_2$ and $z_1$. These results are again computed for the box shown in Fig. 1.

Finally, we investigated the local impact of the scheme on ozone chemistry by examining changes in both the surface area density of PSCs composed of NAT particles and the $ClONO_2$ (chlorine nitrate) + HCl (hydrochloric acid) reaction. This heterogeneous reaction is crucial as in their gas phase HCl and $ClONO_2$ are very unreactive, and so any chlorine they contain is unable to destroy ozone (Solomon, 1999). However, in the presence of a PSC surface (either solid or liquid) they can react with each other to produce $Cl_2$ (chorine gas), as well as the removal of nitric acid ($HNO_3$) from the atmosphere, resulting in the denitrification of the stratosphere, an effect which allows $Cl_2$ to build up during wintertime. In the spring, the presence of solar ultraviolet radiation splits $Cl_2$ into two chlorine atoms (so-called chlorine activation), which plays an important role in stratospheric ozone depletion (Solomon, 1999). Note that these results are calculated over the region 76°S-64°S and 75°W-55°W, which includes the Antarctic Peninsula but is not the box depicted in Fig. 1. Furthermore, to look at the non-local impacts we examined changes to ozone over the high-latitude Southern Hemisphere, as well as temperature and pressure changes in the lower stratosphere, i.e. the polar vortex. Keeble et al. (2014) previously showed that in the version of UM-UKCA used here that polar ozone depletion can have significant impacts on the polar vortex.

## 3 Results

### 3.1 Comparison with observations

Figure 2 compares the probability distributions of $\Delta BT_{SSO}^-$ and $\Delta BT_{AIRS}^-$ over the Antarctic Peninsula from May to October, showing that both distributions peak at similar values (around -0.5 K for $\Delta BT_{SSO}^-$ and -1 K for $\Delta BT_{AIRS}^-$) but differ in terms of their shape, with $\Delta BT_{SSO}^-$ restricted to a relatively narrow range and a high peak compared to a broader range and lower peak for $\Delta BT_{AIRS}^-$. However, the agreement between the two distributions improves over the lower / large cooling part of the tail, with both showing a lower bound of around -6 K, which is perhaps the region of the distribution that is critical for decreasing temperatures below the threshold for PSC formation. Note that a possible reason for the discrepancies between the two





distributions could be that the parameterised results only represent mountain-wave-induced disturbances, while AIRS results include contributions from both orographic and non-orographic source regions. It is noteworthy therefore that there is a much

better agreement between the distributions of $\Delta T_{SSO}^-$ and $\Delta T_{RS}^-$ over Rothera Research Station at an altitude of around 20.4 km (Fig. 3), with both distributions showing a relatively narrow range which peaks at a value of around -0.5 K, and the lower / cooling part of the tail extending to around -8 K.

Figure 4 shows maps detailing the location and frequency that instances of $\Delta BT_{SSO}^- < $ -0.1 K and $\Delta BT_{AIRS}^- < $ -0.1 K, i.e. the regions that contribute the most to the probability distributions shown in Fig. 2. The peak source region of the parameterised

values is over the mid-section and highest region (see Fig. 1) of the Antarctic Peninsula, i.e. centered over Alexander Island and Graham Land, which are regions of maximum $\sigma$ (standard deviation of the SSO) in the UM-UKCA model (not shown), as well as a hot-spot of mountain wave activity (Hoffmann et al. 2013). Note that here there are some contributions/waves from regions over the sea, which is due to the smoothness of the UM-UKCA mean orography (due to its relatively coarse resolution), which results in non-zero values of mean orography and associated SSO values over sea points around the

coastline. By contrast, the AIRS-observed values show the peak source region to be more over the northern section of the Antarctic Peninsula, but also the presence of non-orographic source regions, particularly to the north of the Antarctic Peninsula (as discussed earlier as a possible reason for some of the disagreement between the distributions of parametrised and AIRS-observed cooling-phase in Fig. 2).

### 3.2 Impact on minimum temperatures and formation potential of PSCs

The distributions of temperature difference $T - T_{NAT}$ and $T - T_{ice}$ from the perturbation and control runs are shown in Fig. 5 for the combined months of May to October, and reveal that the addition of $\Delta T_{SSO}^-$ to the explicitly resolved synoptic-scale temperature $T_{UM-UKCA}$ (i.e. $T = T_{UM-UKCA} + \Delta T_{SSO}^-$) in the perturbation run is particular important for temperatures to drop below $T_{ice}$, as without this the ice frost point is rarely exceeded by more than a few degrees Kelvin. For PSCs composed of water ice particles, the addition of $\Delta T_{SSO}^-$ to the synoptic-scale temperature in the perturbation run extends the lower bound of

the distribution from around $T - T_{ice} = 0$ K to $T - T_{ice} = -10$ K, while for PSCs composed of NAT particles it is extended from around $T - T_{NAT} = -10$ K to $T - T_{NAT} = -20$ K. Figure 6 is analogous to Fig. 5, but comparing the distributions of the temperature differences $T - T_{NAT}$ and $T - T_{ice}$ for the individual months of May to October for the perturbation and control runs, indicating that the additional cooling $\Delta T_{SSO}^-$ in the perturbation run is vital if $T$ is to drop below $T_{ice}$ during the months of May, June, September, and October  - as during these months in the control run $T = T_{UM-UKCA}$ alone is too warm, i.e. late

austral autumn / early austral winter, as well as early austral spring (consistent with the findings of McDonald et al. (2009)). Note however that during July and August that the cold side of the tail extends to $T - T_{ice} < 0$ K in the control run using $T = T_{UM-UKCA}$. For PSCs composed of NAT particles the impact of the parameterisation in the perturbation run is particularly important for October (and to a lesser degree September), as this is the only month that the additional cooling $\Delta T_{SSO}^-$ is required for $T$ to drop below $T_{NAT}$, increasing the likelihood of PSC formation in early austral spring. However, it should be noted that



the impact on PSC formation is also dependent on the local mixing ratios of $HNO_3$ and $H_2O$, which in part are affected by PSC formation and sedimentation earlier in the winter. We explore the impacts of the parameterisation on PSC surface area density in section 3.4.

Using Eq. (1), Fig. 7 shows the FP for PSCs composed of both water ice and NAT particles at an altitude of 20.4 km for the individual months from May to October from both the perturbation and control runs. This shows that the FP of PSCs composed

of NAT particles is around two orders of magnitude larger than that for PSCs composed of water ice particles due to them having a higher threshold temperature for formation (i.e. roughly around 195 K for $T_{NAT}$ and 188 K for $T_{ice}$ at this altitude) and hence much more likely that the threshold is exceeded (c.f. Figs. 5 and 6). The results show FP values for NAT particles peaking at around $-4 \times 10^6$ K km$^2$ in June and July, but with little sensitivity in any of the months to the inclusion of the additional cooling $\Delta T_{SSO}^-$ in the perturbation run. However, consistent with Fig. 6 is that the FP of PSCs composed of water ice

particles is highly sensitive to the inclusion of the additional cooling $\Delta T_{SSO}^-$ in the perturbation run, with FP values around 4-5 times larger in July and August if the additional cooling is included compared to it being neglected in the control run, as well as significant increases also occurring during June and September, which otherwise show a negligible FP for the control run. For PSC composed of NAT particles, the FP values obtained from the perturbation and control run are much more similar (c.f. Figs. 5 and 6), although the inclusion of the added cooling in the perturbation run does still result in increases. To further

understand this, Fig. 8 shows maps of the difference in FP between the perturbed and control run for the two types of PSCs examined, revealing that the differences evident in Fig. 7 (i.e. due to the addition of $\Delta T_{SSO}^-$ to the synoptic-scale temperature) are dominated by the contribution from mountain waves originating from the high-elevation base of the Antarctic Peninsula (which Hoffmann et al. (2013) showed was a hot-spot of mountain wave activity).

**3.3 Conditions required for large localised negative temperature anomalies**

Using Eq. (2), Fig. 9 compares the range of vertical wind shear α that was associated with the top 10% (i.e. most cold) and bottom 10% (i.e. least cold) of the distribution of the cooling-phase $\Delta T_{SSO}^-$ at an altitude of 20.4 km. This shows that the largest negative cooling-phases are associated with larger (positive) values of α, which is consistent with the understanding that waves with long vertical wavelengths in the stratosphere generate large temperature fluctuations and are associated with conditions where wind speed increases with height, i.e. causing wave refraction (Wu and Eckermann, 2008). Hoffman et al. (2017) also

showed that such conditions were conducive for the propagation of gravity waves into the lower stratosphere with long vertical wavelengths, which AIRS can best identify. Note that the top 10% and bottom 10% of the distribution was comparatively insensitive to the change in wind direction with height (not shown), which perhaps reflects that the wind regime is predominately unidirectional with height, i.e. a similar structure at many height levels in both the troposphere and lower stratosphere, consistent with an equivalent barotropic structure (Thompson and Wallace, 2000).





### 3.4 Impact on chlorine activation and PSCs over the Antarctic Peninsula

The impact of the additional cooling $\Delta T_{SSO}^{-}$ in the perturbation run on PSCs composed of NAT particles and chlorine activation is shown in Fig. 10. In the control run, the maximum surface area density of PSCs composed of NAT particles is modelled in June at an altitude of around 20 km, and extending from around 10 to 30 km. Between June and September, the surface area of the NAT particles decreases due to both rising (synoptic scale) temperatures and the effects of denitrification and dehydration of the polar vortex by PSC sedimentation (Fahey et al., 1990; Teitelbaum et al., 2001). The result is that by August and September, little PSC surface area remains for chlorine activation. However, in the perturbation run, the surface area density of the NAT particles is increased at higher altitudes throughout the winter and early spring and reduced at lower altitudes. Importantly for chlorine activation in the late winter and spring (August and September), surface area density is increased by up to 20%. Also shown in Fig. 10 is the flux through the $ClONO_2 + HCl$ heterogeneous reaction, a key reaction for the activation of chlorine from the major chlorine reservoir species. Surface area density changes of the NAT particles have only a modest impact on chlorine activation throughout the winter, but the small increases in surface area density in the late winter and early spring in the perturbation experiment result in large increases in chlorine activation throughout August and September, and thus enhancing ozone depletion (Solomon, 1999).

### 3.5 Impact on mean total column ozone, temperature, and pressure over the high-latitude Southern Hemisphere

Figure 11 shows the impact of the additional cooling in the perturbation run on October monthly mean total column ozone. While Fig. 10 highlights the local impacts of the parameterisation scheme on PSC formation and chlorine activation, it can be seen from Fig. 11 that the impacts of the parameterisation scheme extend far beyond the region of the Antarctic Peninsula. This is unsurprising, as not only is the Antarctic Peninsula responsible for differences both upstream and downstream of the region, but other hot-spots of mountain wave activity exist over Antarctica that can also play a role in PSC formation, such as the Transantarctic mountains (e.g. Noel et al., 2009; Hoffmann et al., 2013, 2017; Alexander et al., 2017) which would also be sources of cooling via the parameterisation scheme. While perhaps it would be expected that October monthly mean total column ozone would be reduced above and downwind from the Antarctic Peninsula when the additional cooling $\Delta T_{SSO}^{-}$ is included in the perturbation run, there is little change to column ozone values here. Instead, total column ozone is reduced between 30°E-130°E and increased between 120°W-180°W. This is indicative of a shift of the polar vortex away from the Pacific sector of the Southern Ocean, and towards the Indian Ocean sector. This result is supported by the 25 km pressure and temperature differences between the two simulations, which both indicate a change in the position of the polar vortex (Fig. 11).

### 4 Conclusions

Mountain-wave-induced PSC formation, which is a significant influence on ozone chemistry, is missing from current coarse-resolution global chemistry-climate models because the small-scale temperature fluctuations associated with mountain waves



are neither resolved nor parameterised – limiting our ability to make accurate predictions of stratospheric ozone. This study examines in detail an attempt to make global chemistry-climate models more physically based / comprehensive by including a novel parameterisation of mountain-wave-induced temperature fluctuations inserted into a 30-year run of the global chemistry-climate configuration of the UM-UKCA global chemistry-climate model.

The study firstly examined the detailed representation of episodic and localised wintertime stratospheric cooling-phases over the Antarctic Peninsula, secondly the subsequent impact of the cooling-phases on local chlorine activation and PSC formation, and thirdly the impacts of the scheme over the entire high-latitude Southern Hemisphere (i.e. the inclusion of mountain-wave-induced cooling-phases from many other orographic hot-spots, and not just the Antarctic Peninsula) on ozone and the stratospheric polar vortex. The main findings were:

-    The probability distribution of the parameterised cooling-phases are in reasonable agreement with values derived from long-term AIRS brightness temperature measurements $\Delta BT_{AIRS}^-$ (with a possible reason for the discrepancy being that AIRS also includes contributions from non-orographic source regions) and in excellent agreement with values derived from long-term radiosonde temperature soundings $\Delta T_{RS}^-$ from Rothera Research Station situated on the Antarctic Peninsula.

-    In both cases the agreement with the AIRS and radiosonde values was particularly good for the lower / large cooling part of the tail of the distributions, with a lower bound of up to -6 K for $\Delta BT_{SSO}^-$ and $\Delta BT_{AIRS}^-$ and up to -8 K for $\Delta T_{SSO}^-$ and $\Delta T_{RS}^-$ - which is perhaps the region of the distribution that is critical for decreasing temperatures below the threshold for PSC formation.

       -    The addition of $\Delta T_{SSO}^-$ to the resolved/synoptic-scale temperatures in the UM-UKCA model (i.e. $T = T_{UM-UKCA} +$
335         $\Delta T_{SSO}^-$) results in a considerable increase in the number of instances when minimum temperatures fall below $T_{ice}$ during late austral autumn / early austral winter and early austral spring by extending the lower bound of the $T - T_{ice}$ distribution from around $T - T_{ice} = 0$ K to $T - T_{ice} = -10$ K, i.e. without the additional cooling-phase the ice frost point is rarely exceeded in the model by more than a degree Kelvin or so during these periods.

       -    The addition of $\Delta T_{SSO}^-$ extends the lower bound of the $T - T_{NAT}$ distribution from around $T - T_{NAT} = -10$ K to $T -$
340         $T_{NAT} = -20$ K, however it is only during October (and to a lesser degree September) that the additional cooling $\Delta T_{SSO}^-$ is required for $T$ to drop below $T_{NAT}$, i.e. early austral spring (although it should be noted that the impact on PSC formation is also dependent on the local mixing ratios of $HNO_3$ and $H_2O$, which in part is affected by PSC formation and sedimentation earlier in the winter).

       -    Values of the FP of PSCs composed of water ice particles are many times larger if the additional cooling $\Delta T_{SSO}^-$ is
345         included, while for PSCs consisting of NAT particles although the additional cooling resulted in an increase in FP, it was small.




- The addition of $\Delta T_{SSO}^-$ results in an increase in the surface area density of NAT particles throughout the winter and early spring, which is important for chlorine activation – evident in a large increases in the flux through the $ClONO_2$ + HCl reaction throughout August and September.

- Examination of the total column ozone during October shows that the addition of $\Delta T_{SSO}^-$ results in a reduction between 30°E-130°E and an increase between 120°W-180°W, indicative of a shift of the polar vortex away from the Pacific sector of the Southern Ocean, and towards the Indian Ocean sector.

Note that Keeble et al. (2014) demonstrated that in the version of UM-UKCA used here that polar ozone depletion can have significant impacts on the polar vortex, affecting both the strength and latitude of the westerly polar jet, and this relationship

has also been noted by other studies (e.g. McLandress et al., 2010; Son et al., 2010; Polvani et al., 2011). The study thus shows that both the local and non-local impacts of including the scheme are substantial, and that inclusion of the scheme in a global chemistry-climate model is a step towards it becoming more consistent with our physically based understanding of the atmosphere. This we suggest is essential for understanding how models respond to changes to ozone-depleting substances and greenhouse gases, and hence for improving predictions of ozone and the high-latitude Southern Hemisphere climate system.

Note also that next generation models, such as the ICON-ART (ICOsahedral Nonhydrostatic model with Aerosols and Reactive Trace gases) global modelling system (Schröter et al., 2018), may be able to employ variable spatial resolution with local grid refinement where the resolution increases locally over mountainous regions so that the mountain-wave-induced temperature fluctuations are resolved explicitly, negating the need for their parameterisation.

As one of the main aims of global chemistry-climate models is the prediction of ozone, which to determine accurately requires

a realistic treatment of PSCs, further work will focus on assessing the representation of PSCs in this state-of-the-art configuration of the UM-UKCA by comparing results in both hemispheres against a comprehensive climatology of PSC coverage based on MIPAS (Michelson Interferometer for Passive Atmospheric Sounding) observations (Spang et al., 2018). Moreover, although the UM-UKCA model (in common with many other global climate models) employs a rather simplistic PSC scheme which limits its ability to accurately predict ozone, the improved representation of PSC formation detailed in this

study will also eventually be used to develop better projections of future polar ozone levels in response to climate change, such as narrowing uncertainties in the rate and timing of the closure of the Antarctic ozone hole (Eyring et al., 2013).

**Code/data availability**

The AIRS measurements of brightness temperature perturbations used in this study are registered under https://www.re3data.org/repository/r3d100012430, with a DOI http://doi.org/10.17616/R34J42, and can be downloaded from

https://datapub.fz-juelich.de/slcs/airs/gravity_wave. The high-resolution radiosonde data from Rothera Research Station can be downloaded from https://catalogue.ceda.ac.uk/uuid/37f2bef57e28bcd780a5cbfe077f4bf8. Please contact the lead author if you would like access to the UM-UKCA output. Data analysis in this paper was conducted using the open source python libraries SciTools-Iris (https://scitools.org.uk/iris) and Pandas (https://pandas.pydata.org/).



**Author contributions**

AO, SH and AD conceived and worked on the research project, with AD undertaking much of the analysis for this study during a 4-month internship at the British Antarctic Survey. SH subsequently revised many of the plots included in the manuscript. AO implemented the parameterisation scheme into the UM-UKCA model, and also ran the model simulations. AO also wrote the majority of the paper. Many of the ideas for the work were generated during a mini-workshop at the British Antarctic Survey, which was attended by all the authors. LH provided the AIRS-based observations, and expertise on how they should

be utilised. RS provided expert advice on the calculation of temperature thresholds for PSC formation. TMG provided the radiosonde data, and advice on how it should be used. JK produced the plots investigating local PSC formation and heterogeneous chemistry, as well the non-local impacts of the scheme by examining changes to ozone as well as the position of the polar vortex. NLA provided expert advice on the setting up and running of the UM-UKCA simulations. Additionally, all authors contributed to the interpretation and writing of the paper.

**Competing interests**

The authors declare that they have no conflict of interest.

**Acknowledgements**

This work was undertaken as part of the Polar Science for Planet Earth Programme of the British Antarctic Survey and funded by the Natural Environment Research Council (NERC). JK and NLA thank NERC through NCAS for financial support. This

work used the ARCHER UK National Supercomputing Service. We acknowledge the hard work of the Rothera field staff, who made the radiosonde measurements

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


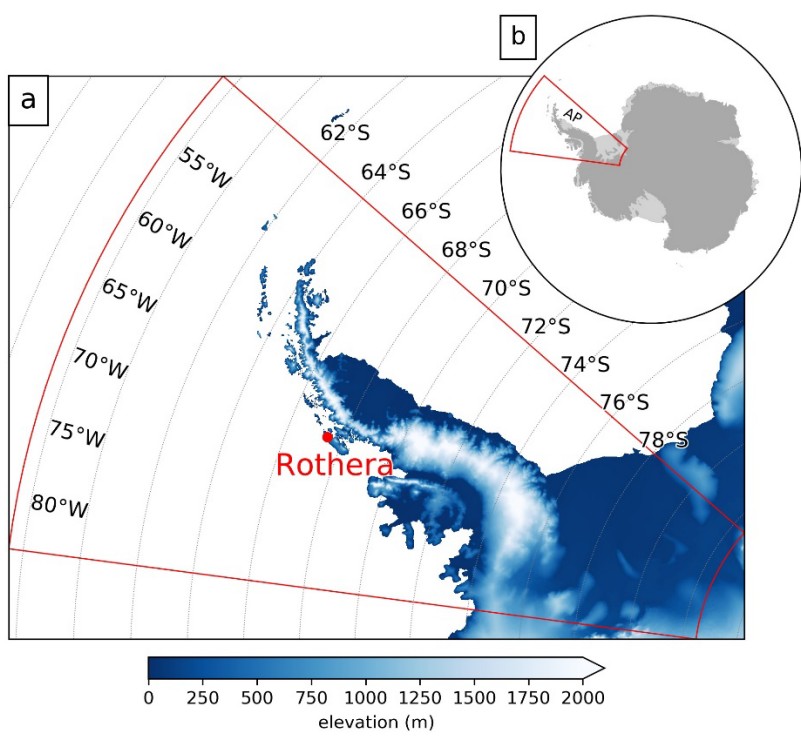


**Figure 1: Maps of the (a) Antarctic Peninsula region showing the box used to compute both the parameterised and AIRS results, as well as the location of Rothera Research Station where the radiosondes are launched and the elevation of the orography, and (b) Antarctica, with the locations of both the box and the Antarctic Peninsula (AP) indicated. Note that orography dataset used in panel (a) is Bedmap2 (Fretwell et al., 2013).**



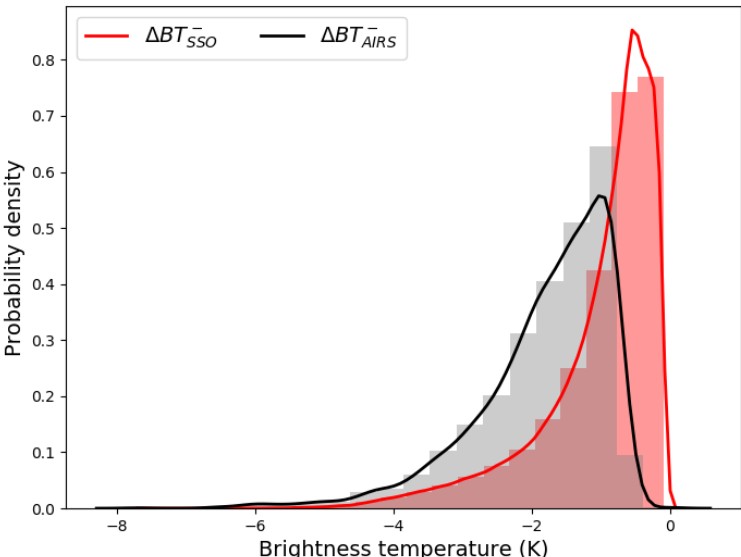

**Figure 2: Comparison of the probability distribution of brightness temperature perturbations (K) due to mountain-wave-induced stratospheric cooling over the Antarctic Peninsula between the parameterisation scheme $\Delta BT_{SSO}^{-}$ (red line) and the AIRS observations $\Delta BT_{AIRS}^{-}$ (black line) for May to October. The AIRS values are from the 666.5 cm$^{-1}$ radiance channel, for a 16-yr period from 2002 to 2017 (and includes some contribution from non-orographic wave sources). The parameterised values are the weighted sum of $\Delta T_{SSO}^{-}$ from the perturbation run over all vertical model levels from 15 to 45 km (using the AIRS kernel function for the**
**666.5 cm$^{-1}$ radiance channel), which is required to convert $\Delta T_{SSO}^{-}$ to $\Delta BT_{SSO}^{-}$. Note that a minimum threshold of BT < -0.1 K is used to reduce the inclusion of noise/spurious events. Both the parameterised and AIRS results are computed within the box indicated in Figure 1.**





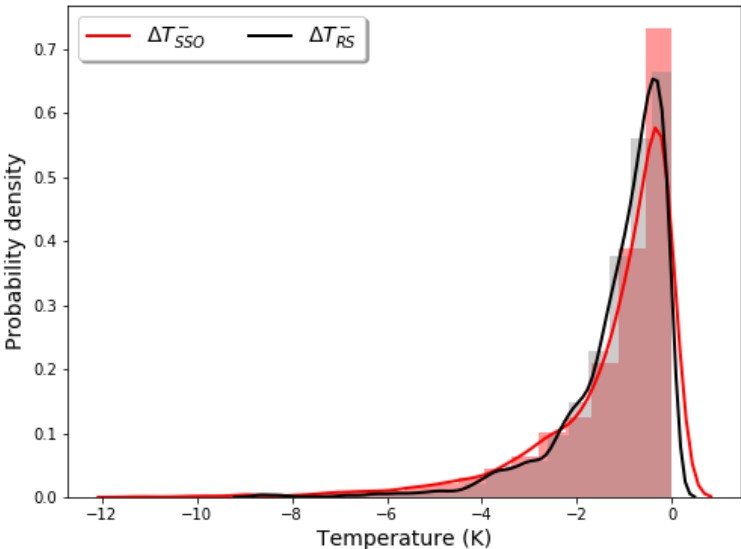

**Figure 3: Comparison of the probability distribution of the temperature perturbations (K) due to mountain-wave-induced cooling over Rothera Research Station on the Antarctic Peninsula between the parameterisation scheme $\Delta T^-_{SSO}$ (red line) and the radiosonde observations $\Delta T^-_{RS}$ (black line) for May to October at an altitude of around 20.4 km. The radiosondes are launched around two to four times a week from Rothera for a 14-yr period from 2002 to 2015 (see Fig. 1 for location) and compared against parameterised values from the perturbation run, which are taken from the grid box that contains this location. Note that a minimum threshold of T < -0.1 K is used to reduce the inclusion of noise/spurious events.**






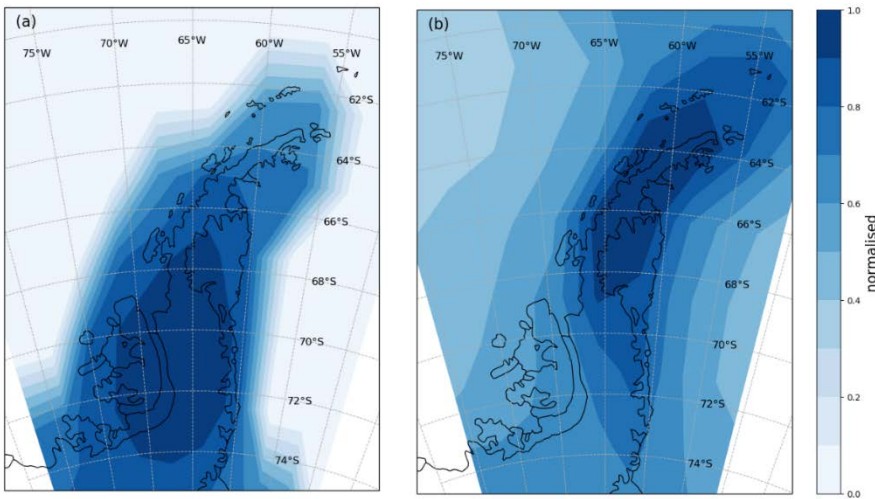

**Figure 4: Map of the normalised number of instances that mountain-wave-induced cooling occurs over the Antarctic Peninsula in the (a) parameterised and (b) AIRS observations for May to October. Both the parameterised and AIRS results are based on the same information used to produce the probability distributions in Fig. 2. Note that the AIRS results also include some contribution from non-orographic wave sources. Note also that the maximum number is used to rescale/normalise the values from 0 to 1.**








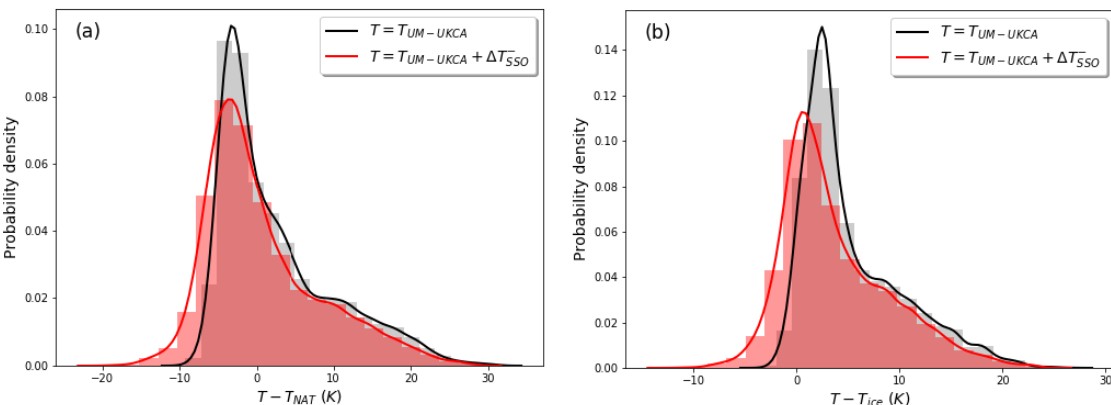

**Figure 5: Comparison of the probability distributions of the temperature differences (K) for (a) $T - T_{NAT}$ and (b) $T - T_{ice}$ over the Antarctic Peninsula at an altitude of 20.4 km for the combined months of May to October for the perturbation run (red line) and the control run (black line), i.e. for $T$ equal to either $T_{UM-UKCA} + \Delta T^-_{SSO}$ (perturbation run with the parameterisation scheme on) or $T$ equal to $T_{UM-UKCA}$ (control run with the parameterisation scheme off). Note that the results are computed within the box indicated in Fig. 1.**





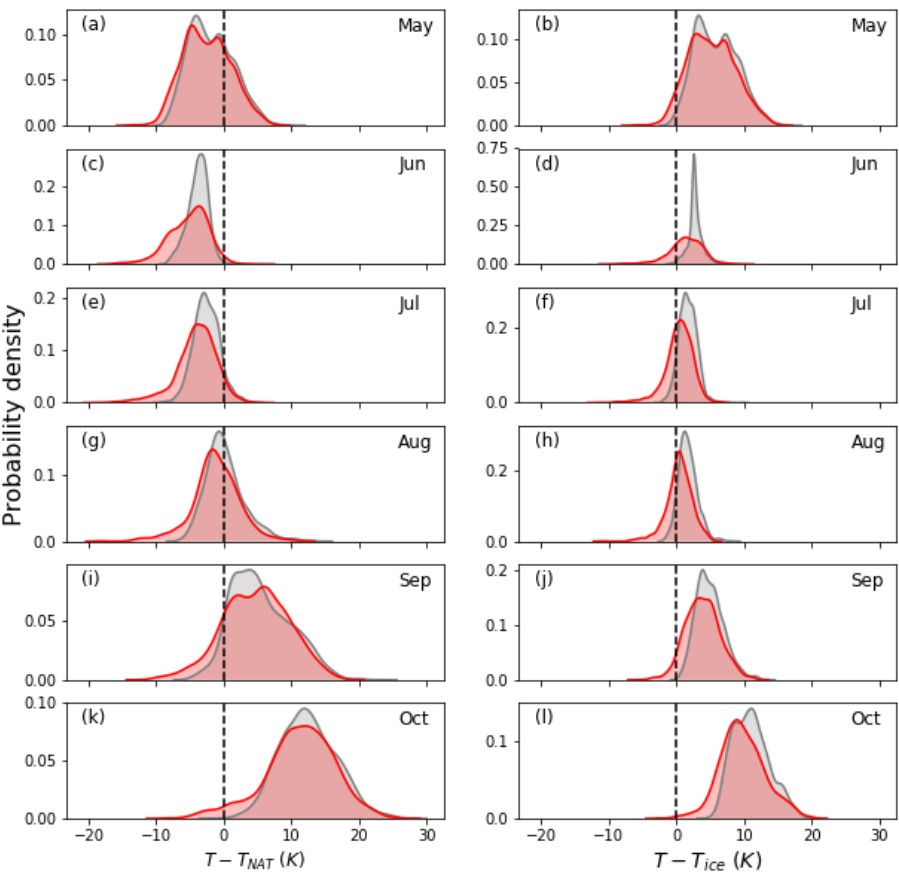

**Figure 6: As Fig. 5, but for the individual months from May (top row) to October (bottom row), with the panels on the left-hand-side showing results for PSCs composed of NAT particles ($T - T_{NAT}$), and on the right-hand-side for PSCs composed of water ice particles ($T - T_{ice}$) at an altitude of 20.4 km. Results are shown for the perturbation run (red line) and the control run (black line), i.e. for $T$ equal to either $T_{UM-UKCA} + \Delta T_{SSO}^-$ (perturbation run with the parameterisation scheme on) or for $T$ equal to $T_{UM-UKCA}$ (control run with the parameterisation scheme off). The vertical dashed line denotes either $T - T_{NAT} = 0$ or $T - T_{ice} = 0$. Note that the results are computed within the box indicated in Fig. 1.**





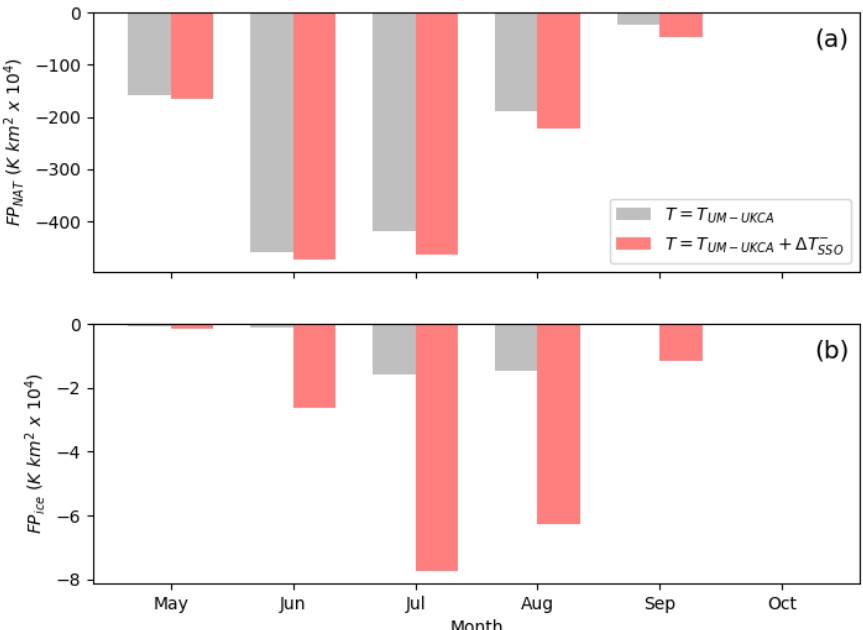

**Figure 7: Histograms showing the monthly mean formation potential ($\times 10^4$ K km$^2$; see Equation (1) for definition) for PSCs made**
**from (a) NAT and (b) ice particles during each individual month from May to October at an altitude of 20.4 km for the perturbation**
**run (red) and the control run (grey), i.e. for $T$ equal to either $T_{UM-UKCA} + \Delta T_{SSO}^-$ (perturbation run with the parameterisation**
**scheme on) or $T$ equal to $T_{UM-UKCA}$ (control run with the parameterisation scheme off). Note that the results are computed within**
**the box indicated in Fig. 1.**







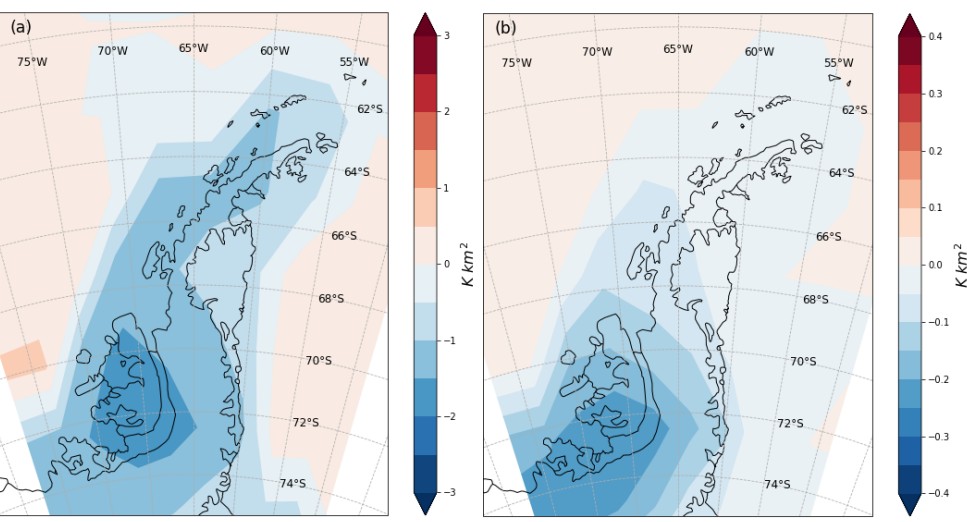

**Figure 8: Maps of the differences in mean monthly FP (K km$^2$) between the perturbation run and the control run for the combined months of May to October over the Antarctic Peninsula at an altitude of 20.4 km for PSCs composed of (a) NAT and (b) ice particles, i.e. the difference between using $T$ equal to either $T_{UM-UKCA} + \Delta T_{SSO}^-$ (perturbation run with the parameterisation scheme on) and $T$ equal $T_{UM-UKCA}$ (control run with the parameterisation scheme off).**





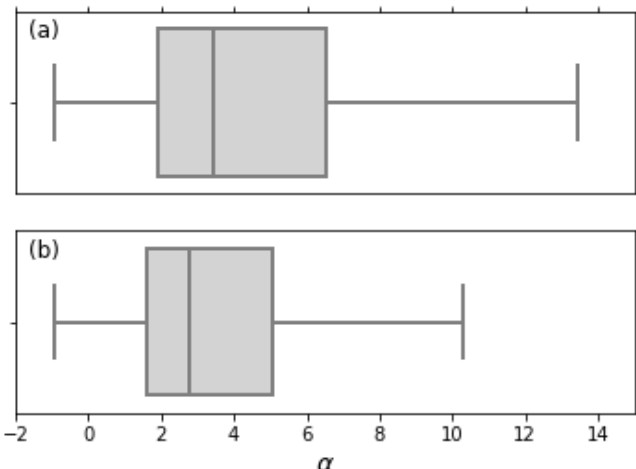

**Figure 9: Box and whiskers plot showing the range of the vertical wind shear α (see Eq. 2 for definition) for the (a) top 10% and (b) bottom 10% of the probability distribution of the parameterised cooling-phase $\Delta T^-_{SSO}$ over the Antarctic Peninsula from the perturbed run for May to October at an altitude of 20.4 km, i.e. the most cold (top 10%) and least cold (bottom 10%) of the cooling-phase events. Note that the results are computed within the box indicated in Fig. 1.**








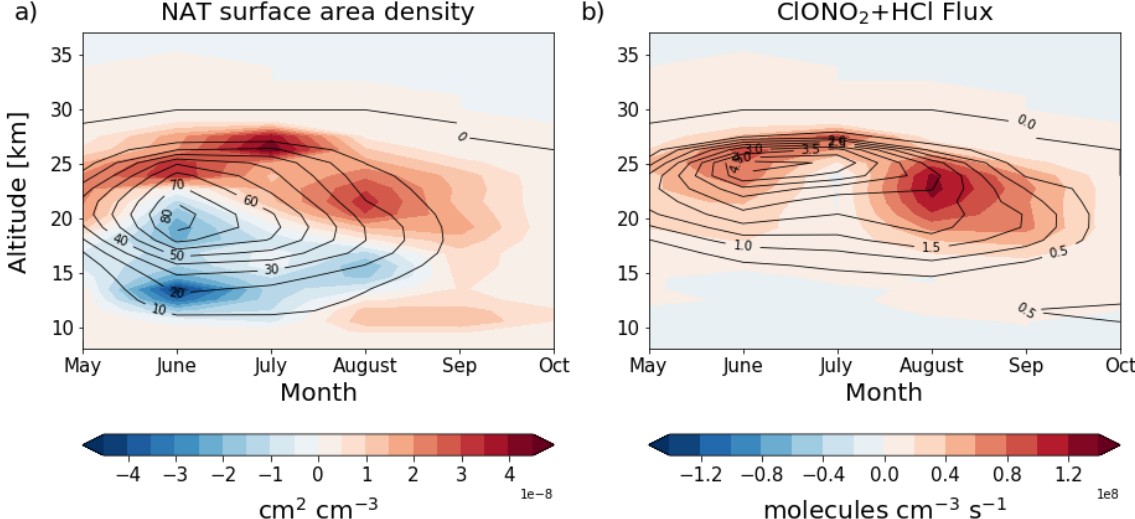

**Figure 10: Altitude versus time plots of the differences in (a) NAT PSC surface area density ($\times 10^{-8}$ cm$^2$ cm$^{-3}$; shading) and (b) the flux through the ClONO$_2$ + HCl reaction ($\times 10^8$ molecules cm$^{-3}$ s$^{-1}$; shading) between the perturbed run and the control run, averaged over the Antarctica Peninsula (over the region 76°S-64°S and 75°W-55°W), i.e. the difference between using $T$ equal to either $T_{UM-UKCA} + \Delta T_{SSO}$ (perturbation run with the parameterisation scheme on) and $T$ equal to $T_{UM-UKCA}$ (control run with the parameterisation scheme off). Also shown in each panel are the respective values from the control simulation using a value of $T$ equal to $T_{UM-UKCA}$ (contours).**







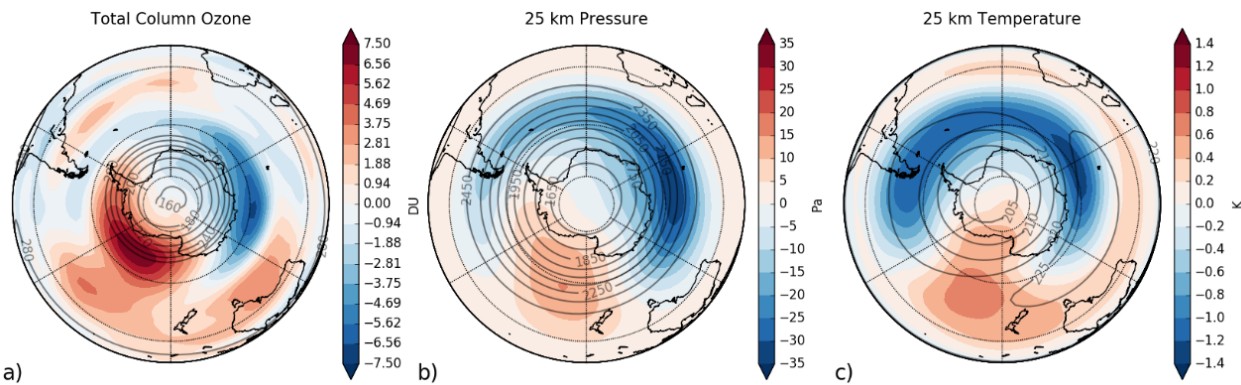

**Figure 11: Maps of the average differences in October monthly mean total column ozone (a; units of DU), 25 km pressure (b, units of Pa) and 25 km temperature (c, units of K) between the perturbed run and the control run, i.e. the difference between using $T$ equal to either $T_{UM-UKCA} + \Delta T^-_{SSO}$ (perturbation run with the parameterisation scheme on) and $T$ equal to $T_{UM-UKCA}$ (control run with the parameterisation scheme off). Also shown in each panel are the respective values from the control simulation using a value of $T$ equal to $T_{UM-UKCA}$ (contours).**