# Peer review of "PSCs initiated by mountain waves in a global chemistry-climate model: A missing piece in fully modelling polar stratospheric ozone depletion"

_Atmospheric Chemistry and Physics, 2020_

## Referee Comment (RC1) · Anonymous Referee #1 · 12 Aug 2020

The paper is a self-contained study of the impact of mesoscale mountain wave-induced temperature fluctuations on the formation of polar stratospheric clouds (PSCs) over the Antarctic Peninsula. A simple parametrization of these episodic and localized wintertime stratospheric negative temperature fluctuations is implemented into the global chemistry-climate configuration of the UM-UKCA (Unified Model - United Kingdom Chemistry and Aerosol) model for two 30-year runs, a perturbation simulation with and a control run without parametrized temperature perturbations. Meaningful results are presented in form of probability distributions of the parameterized cooling-phases which are compared to AIRS observations and high-resolution radiosonde temperature soundings from the Rothera research station. Probably the main result of this study is that adding the parameterized, mountain wave-induced cooling to the resolved temperatures in the UM-UKCA model results in a considerable increase in the number of instances when minimum temperatures fall below the threshold temperatures necessary for the formation of PSCs in the transitional seasons of fall and spring.

The paper is clearly written, contains all information needed to follow the narrative, and the 10 figures are precise and easy readable. Although one could ask oneself why parametrizations like the one presented are necessary in the age of increasing computer power and finer spatial and temporal resolutions of general circulation models, the quantification of mesoscale temperature fluctuations and the evaluation on their impact on stratospheric chemistry are valuable contributions to our knowledge. I have only a few, really minor remarks that could be considered in the revised version of this paper.

**line 57**: The earliest paper stating the impact of mesoscale mountain waves on the PSC formation is by Ken Carslaw et al. and is worth mentioning here:

Carslaw, Ken et al. (1998). Increased stratospheric ozone depletion due to mountain-induced atmospheric waves. Nature, v.391, 675-678 (1998). 391. 10.1038/35589.

**line 107-109**: A very supporting paper about the blocking is the early GRL publication by Julio Bacmeister. I think, this contribution was the origin of all considerations using blocking heights in parametrizations. A paper also worth citing, especially, as it considers the same geographical region:

Bacmeister, J. et al., 1990: ER-2 mountain wave encounter over Antarctica: Evidence for blocking, GRL, https://doi.org/10.1029/GL017i001p00081

**line 139**: The Hersbach et al. ERA5 paper is now online published at the QJ web site: https://rmets.onlinelibrary.wiley.com/doi/10.1002/qj.3803

**line 147**: replace "km" by "km$^2$" twice

**line 154**: Hoffmann with double "nn" at the end

**line 224, 235**: One dominant characteristics of the mountain wave propagation from the Antarctic peninsula is the frequent occurrence of oblique ray paths. This process might lead to a horizontal shift of mountain wave activity away from the actual sources towards the Drake Passage. I believe, the difference between the AIRS results and the simple, linear mountain wave parametrization as shown in Figure 4 can be easily explained by this process without referring to non-orographic sources. If you hesitate to investigate this aspect in more detail I would at least suggest adding suitable references that point to these well-known processes:

Preusse, P. et al., 2002: Space-based measurements of stratospheric mountain waves by CRISTA, 1. Sensitivity, analysis method, and a case study. J. Geophys. Res., 107(D23), 8178, doi:10.1029/2001JD000699.

Sato, K. et al. (2012).Gravity wave characteristics in the southern hemisphere revealed by a high-resolution middle-atmosphere general circulation model. Journal of the Atmospheric Sciences,69(4), 1378-1396.  doi:  10.1175/JAS-D-11-0101.1

There are more papers that also document the oblique mountain wave propagation into the polar night jet that are based on recent results from the DEEPWAVE campaign, e.g.

Ehard, B., et al  (2017).  Horizontal propagation of large-amplitude mountain waves into the polar night jet. Journal of Geophysical Research:  Atmospheres, 122(3), 1423-1436.  doi:  10.1002/2016JD025621

Jiang, Q., et al.  (2019).   Stratospheric trailing gravity waves from New Zealand. Journal of the Atmospheric Sciences, 76(6), 1565-1586.  doi:  10.1175/JAS-D-18-0290.1

line 279: I would add " … towards longer vertical wavelengths" before the parenthesis " (Wu ....)". Additionally, the shear calculation by Eq. 2 neglects recent results by the DEEPWAVE campaign about the existence of the "valve" layer:

Kruse, C. G. et al. 2016: The Midlatitude Lower-Stratospheric Mountain Wave "Valve Layer". J. Atmos. Sci., 73, 5081–5100, https://doi.org/10.1175/JAS-D-16-0173.1.

Indeed, how the waves propagate into the stratosphere (vertically as well as horizontally) depends largely and critically on the wind profile as nicely documented by

Bramberger, M., et al. (2017). Does Strong tropospheric forcing cause large-amplitude mesospheric gravity waves? A DEEPWAVE case study. Journal of Geophysical Research: Atmospheres, 122, 11,422– 11,443. https://doi.org/10.1002/2017JD027371

---

## Referee Comment (RC2) · Anonymous Referee #2 · 17 Aug 2020

Summary:

Orr et al. implement a mountain-wave parameterisation scheme within the UM-UKCA GCM, in its chemistry-climate mode. They use a 30-year run (perpetual 2000) to investigate the regional (over the Antarctic Peninsula) effects on PSC formation throughout the winter by quantifying the wave-induced negative temperature perturbations and evaluating against AIRS satellite and Rothera radiosondes. The authors demonstrate that inclusion of the parameterisation scheme enables substantially more PSC formation, especially for the ice class, whose formation potential is particularly sensitive to

changes in temperature at these relatively northerly latitudes. Orr et al. then investigate the Antarctic-wide effects of inclusion of their parameterisation scheme and show the differences to the total column ozone distribution during October. The authors demonstrate the value of inclusion of this parameterisation in the UM-UKCA in their final sections (Sec 3.5 and 3.6), which quantify changes in NAT surface area density and the ClONO2 + HCl flux (Fig 10) and the differences in October monthly mean column ozone, pressure, and temperature (Fig 11).

I have only a few minor comments (and a few technical points) for this manuscript and I recommend its publication once the authors have considered the below points.

Minor Comments:

Line 31, and at other locations throughout the manuscript: You use the phrase 'ice frost point is rarely exceeded' here. But really, you mean that temperatures rarely fall below the ice frost point temperature (without the additional wave-induced cooling phase). I suggest you rewrite these phrases throughout the manuscript to make this clear, and avoid the phrase 'rarely exceeded'.

Line 222: Comments that the lowest temperatures (around -6K) are 'perhaps the region of the distribution that is critical for decreasing below the threshold for PSC formation'. This would probably only be true early in the season and at the end, when synoptically the temperatures are warm. On the other hand, in mid-winter, you would be more likely to only need a small negative perturbation to form (ice) PSC. Relatedly, Figures 2 & 3 indicate that very few waves have large negative perturbations, but that is not of course saying these large amplitudes are unimportant especially at season start and end – perhaps this point could be made too.

Line 225: Comparison of parameterised perturbation and Rothera perturbations. A clear agreement. However, all observations are only seeing part of the wave spectrum, and in particular radiosondes are preferentially observing inertia-gravity waves. Moffat-Griffen et al. (JGR 2011, doi 10.1029/2010JD015349) concluded that winter & spring

radiosonde observations of stratospheric waves at Rothera may also be in part due to non-orographic sources (e.g. vortex edge). You mention in the text that AIRS includes orographic & non-orographic wave sources. It seems to me that you should note that the Rothera radiosondes likely do so too.

Line 232: A 'hot-spot' within the Antarctic Peninsula 'hot-spot'?

Line 245: You write that the lower bound of the T-Tice distribution is around 0K in the control case, but it seems about -2K or even -3K to me (black line in Figure 5b). Please check. Maybe a better way (more quantitative way) of comparing is to say at what temperature the e.g. 1% limit is at?

Line 332: See comment from line 222 above.

Technical Comments:

Lines 26 & 27: I don't think 'AIRS-observations' and 'radiosonde-observations' need hyphens

Line 69: 'typically have horizontal wavelengths'

Line 112 & following: Subscript the '3' and '2' in HNO3, H2O

Line 136: suggest rewording to 'at pressure heights around 850 hPa'

Line 139: I think the reference for ERA5 is Hersbach et al, QJRMS 2020, doi:10.1002/qj.3803

Line 139 & 140: You use 'shown' and 'not shown' in the same sentence, please reword

Line 236: Sentence seems incomplete.

Line 272 'high-altitude', not 'high-elevation'

Line 340. Phrase 'only during October (and to a lesser degree September)' doesn't make sense. Reword.

---

## Author Comment (AC1) · 17 Sep 2020

Author Comment

Reviewer #1

We are grateful for the Reviewers insightful comments. We are pleased that they found the results meaningful, the paper clearly written, and the figures precise and readable. We note that the Reviewer has only suggested some minor remarks, which we have answered below.

[Figure]

(1) line 57: The earliest paper stating the impact of mesoscale mountain waves on the PSC formation is by Ken Carslaw et al. and is worth mentioning here: Carslaw, Ken et al. (1998). Increased stratospheric ozone depletion due to mountain-induced atmospheric waves. Nature, v.391, 675-678 (1998). 391. 10.1038/35589.

Reply: This change has been made and we have added the Carslaw et al. (1998) reference.

(2) Comment - line 107-109: A very supporting paper about the blocking is the early GRL publication by Julio Bacmeister. I think, this contribution was the origin of all considerations using blocking heights in parametrizations. A paper also worth citing, especially, as it considers the same geographical region:

Bacmeister, J. et al., 1990: ER-2 mountain wave encounter over Antarctica: Evidence for blocking, GRL, https://doi.org/10.1029/GL017i001p00081

Reply: This suggestion has been followed. The description of the parameterisation scheme has been revised to include the text: 'The scheme also includes the effects of low-level flow blocking (Bacmeister et al., 1990), such that the initial wave amplitude is set equal to the "effective" mountain height . . ..'

(3) Comment - line 139: The Hersbach et al. ERA5 paper is now online published at the QJ web site: https://rmets.onlinelibrary.wiley.com/doi/10.1002/qj.3803

Reply: This change has been made and we have added the Hersbach et al. (2020) reference.

(4) Comment - line 147: replace "km" by "km2" twice

Reply: This change has not been made. The original description of the AIRS specification supplied by NASA (https://airs.jpl.nasa.gov/mission/instrument/specs/) refers to e.g. 41 $\times$ 21.4 km, so we have not implemented the suggestion to replace 'km' with 'km2'.

(5) Comment - line 154: Hoffmann with double "nn" at the end

Reply: This change has been made. There were a number of occasions where Hoffmann was wrongly spelt, i.e. Hoffman. These have all been corrected.

(6) Comment - line 224, 235: One dominant characteristics of the mountain wave propagation from the Antarctic peninsula is the frequent occurrence of oblique ray paths. This process might lead to a horizontal shift of mountain wave activity away from the actual sources towards the Drake Passage. I believe, the difference between the AIRS results and the simple, linear mountain wave parametrization as shown in Figure 4 can be easily explained by this process without referring to non-orographic sources. If you hesitate to investigate this aspect in more detail I would at least suggest adding suitable references that point to these well-known processes:

Preusse, P. et al., 2002: Space-based measurements of stratospheric mountain waves by CRISTA, 1. Sensitivity, analysis method, and a case study. J. Geophys. Res., 107(D23), 8178, doi:10.1029/2001JD000699.

Sato, K. et al. (2012). Gravity wave characteristics in the southern hemisphere revealed by a high-resolution middle-atmosphere general circulation model. Journal of the Atmospheric Sciences,69(4), 1378-1396. doi: 10.1175/JAS-D-11-0101.1

There are more papers that also document the oblique mountain wave propagation into the polar night jet that are based on recent results from the DEEPWAVE campaign, e.g.

Ehard, B., et al (2017). Horizontal propagation of large-amplitude mountain waves into the polar night jet. Journal of Geophysical Research: Atmospheres, 122(3), 1423-1436. doi: 10.1002/2016JD025621

Jiang, Q., et al. (2019). Stratospheric trailing gravity waves from New Zealand. Journal of the Atmospheric Sciences, 76(6), 1565-1586. doi: 10.1175/JAS-D-18-0290.1

Reply: This suggestion has been followed. The explanation for the disparity in Figure 4 in the revised manuscript has been revised to include the text '... (b) the (vertical-only

propagation) parameterisation scheme does not represent the horizontal propagation of waves (Preusse et al., 2002; Sato et al., 2012), which could be potentially important here and result in a horizontal shift of mountain wave activity away from the source region.'

(7) Comment - line 279: I would add " ... towards longer vertical wavelengths" before the parenthesis"(Wu ....)". Additionally, the shear calculation by Eq. 2 neglects recent results by the DEEPWAVE campaign about the existence of the "valve" layer:

Kruse, C. G. et al. 2016: The Midlatitude Lower-Stratospheric Mountain Wave "Valve Layer". J. Atmos. Sci., 73, 5081–5100, https://doi.org/10.1175/JAS-D-16-0173.1.

Indeed, how the waves propagate into the stratosphere (vertically as well as horizontally) depends largely and critically on the wind profile as nicely documented by

Bramberger, M., et al. (2017). Does Strong tropospheric forcing cause large-amplitude mesospheric gravity waves? A DEEPWAVE case study. Journal of Geophysical Research: Atmospheres, 122, 11,422–11,443. https://doi.org/10.1002/2017JD027371

Reply: These suggestions have been included. We have made two changes to the revised manuscript. Firstly, the suggested text and Bramberger et al citation have been added, and the manuscript now states 'causing wave refraction towards longer vertical wavelengths (Wu and Eckermann, 2008; Bramberger et al., 2017).' Secondly, the following sentence has been added to explain the limitations of Eq. 2: 'Note that this approach would not represent the impact of more local variations in $\alpha$ that also influence vertical propagation (Kruse et al., 2016).'

---

## Author Comment (AC2) · 17 Sep 2020

Author Comment

Reviewer #2

We are grateful for the Reviewers insightful comments and that they are happy to recommend its publication after we have addressed their minor comments, which we have answered below.

[Figure]

(1) Comment - Line 31, and at other locations throughout the manuscript: You use the phrase 'ice frost point is rarely exceeded' here. But really, you mean that temperatures rarely fall below the ice frost point temperature (without the additional wave-induced cooling phase). I suggest you rewrite these phrases throughout the manuscript to make this clear, and avoid the phrase 'rarely exceeded'.

Reply: This suggestion has been followed. The text 'the ice frost point is rarely exceeded above the Antarctic Peninsula' has been revised to 'the temperature rarely falls below the ice frost point temperature above the Antarctic Peninsula'. A similar correction was made a further four times at other locations throughout the manuscript.

(2) Comment - Line 222: Comments that the lowest temperatures (around -6K) are 'perhaps the region of the distribution that is critical for decreasing below the threshold for PSC formation'. This would probably only be true early in the season and at the end, when synoptically the temperatures are warm. On the other hand, in mid-winter, you would be more likely to only need a small negative perturbation to form (ice) PSC. Relatedly, Figures 2 & 3 indicate that very few waves have large negative perturbations, but that is not of course saying these large amplitudes are unimportant especially at season start and end – perhaps this point could be made too.

Reply: These suggestions have been followed. The sentence on line 222 has been modified to state '... showing a lower bound of around -6 K, which is perhaps the region of the distribution that is critical for decreasing temperatures below the threshold for PSC formation (particularly during early winter and early spring).' We have also added an additional sentence to cover the Reviewer's second point: 'Both Figs. 2 and 3 suggest that Antarctic Peninsula mountain waves with relatively large amplitudes of 5-10 K are uncommon (although it is noted that Eckermann et al. (2009) observed waves in this region with an amplitude of around 10 K for a particular case study).'

(3) Comment - Line 225: Comparison of parameterised perturbation and Rothera perturbations. A clear agreement. However, all observations are only seeing part of

the wave spectrum, and in particular radiosondes are preferentially observing inertia-gravity waves. Moffat-Griffin et al. (JGR 2011, doi 10.1029/2010JD015349) concluded that winter & spring radiosonde observations of stratospheric waves at Rothera may also be in part due to non-orographic sources (e.g. vortex edge). You mention in the text that AIRS includes orographic & non-orographic wave sources. It seems to me that you should note that the Rothera radiosondes likely do so too.

Reply: This suggestion has been followed. An additional sentence has been included which says: 'Note that the radiosonde results may also include contributions from non-orographic sources, such as from waves generated by the edge of the polar stratospheric vortex (Moffat-Griffin et al., 2011).'

(4) Comment - Line 232: A 'hot-spot' within the Antarctic Peninsula 'hot-spot'?

Reply: This sentence has been revised by removing the section 'as well as a hot-spot of mountain wave activity (Hoffmann et al. 2013)'.

(5) Comment - Line 245: You write that the lower bound of the T-Tice distribution is around 0K in the control case, but it seems about -2K or even -3K to me (black line in Figure 5b). Please check. Maybe a better way (more quantitative way) of comparing is to say at what temperature the e.g. 1% limit is at?

Reply: This change has been made, and the sentence now states that the difference is from -2 to -3 K.

(6) Comment - Line 332: See comment from line 222 above.

Reply: This suggestion has been followed. As with the reply to the comment on line 222 above, this sentence has been modified to include the phrase '(particularly during early winter and early spring)'.

Technical Comments:

(7) Comment - Lines 26 & 27: I don't think 'AIRS-observations' and 'radiosondeobservations' need hyphens.

Reply: This change has been made.

(8) Comment - Line 69: 'typically have horizontal wavelengths'

Reply: This change has been made.

(9) Comment - Line 112 & following: Subscript the '3' and '2' in HNO3, H2O

Reply: These changes have been made.

(10) Comment - Line 136: suggest rewording to 'at pressure heights around 850 hPa'

Reply: This change has been made.

(11) Comment - Line 139: I think the reference for ERA5 is Hersbach et al, QJRMS 2020, doi:10.1002/qj.3803

Reply: This change has been made.

(12) Comment - Line 139 & 140: You use 'shown' and 'not shown' in the same sentence, please reword

Reply: This suggestion has been followed. This sentence has been reworded to 'To test this here, the 30-year mean wind at 850 hPa for austral winter (June-July-August) from the control experiment was computed and found to be in excellent agreement with the climatological mean from the reanalysis-product ERA5 (i.e. the fifth-generation reanalysis product from ECMWF, Hersbach et al., 2020) over the 1979 to 2019 period (not shown).'

(13) Comment - Line 236: Sentence seems incomplete.

Reply: This suggestion has been followed. The sentence has been revised to 'By contrast, the AIRS-observed values show the peak source region to be more over the northern section of the Antarctic Peninsula. Note that the AIRS-observed values also show some contributions from over the sea surrounding the Peninsula, which

as discussed earlier is a possible reason for some of the disagreement between the distributions of parametrised and AIRS-observed cooling-phase in Fig. 2.'

(14) Comment - Line 272 'high-altitude', not 'high-elevation'

Reply: This change has been made.

(15) Comment - Line 340. Phrase 'only during October (and to a lesser degree September)' doesn't make sense. Reword

Reply: This suggestion has been followed and the sentence rewritten and simplified.

---

## Author Response (AR2)

**This document describes the revised response to a minor comment by Reviewer #1 to '***line 147: replace "km" by "km2" twice'.*

**Author Comments**

**Reviewer #1**

*(4) Comment - line 147: replace "km" by "km2" twice*

**Reply: This change has been made (in line #176 of the revised manuscript, below).**

[revised manuscript text omitted]